# Urinary Tract Involvement in Wolfram Syndrome: A Narrative Review

**DOI:** 10.3390/ijerph182211994

**Published:** 2021-11-15

**Authors:** Alberto La Valle, Gianluca Piccolo, Mohamad Maghnie, Giuseppe d’Annunzio

**Affiliations:** 1Department of Neuroscience, Rehabilitation, Ophthalmology, Genetics, Maternal and Child Health University of Genoa, 16147 Genoa, Italy; albertolavalle88@gmail.com (A.L.V.); giangi.piccolo@gmail.com (G.P.); mohamadmaghnie@gaslini.org (M.M.); 2Pediatric Clinic and Endocrinology Unit, Department of General and Specialist Pediatric Sciences, University of Genoa, 16147 Genoa, Italy; 3Pediatric Clinic and Endocrinology Unit, IRCCS Istituto Giannina Gaslini, 16147 Genoa, Italy

**Keywords:** Wolfram Syndrome, neurological dysfunction, urodynamic, hydroureteronephrosis, bladder dysfunction

## Abstract

Wolfram Syndrome (WS) is a rare neurodegenerative disease with autosomal recessive inheritance and characterized by juvenile onset, non-autoimmune diabetes mellitus and later followed by optic atrophy leading to blindness, diabetes insipidus, hearing loss, and other neurological and endocrine dysfunctions. A wide spectrum of neurodegenerative abnormalities affecting the central nervous system has been described. Among these complications, neurogenic bladder and urodynamic abnormalities also deserve attention. Urinary tract dysfunctions (UTD) up to end stage renal disease are a life-threatening complication of WS patients. Notably, end stage renal disease is reported as one of the most common causes of death among WS patients. UTD have been also reported in affected adolescents. Involvement of the urinary tract occurs in about 90% of affected patients, at a median age of 20 years and with peaks at 13, 21 and 33 years. The aim of our narrative review was to provide an overview of the most important papers regarding urological impairment in Wolfram Syndrome. A comprehensive search on PubMed including Wolfram Syndrome and one or more of the following terms: chronic renal failure, bladder dysfunction, urological aspects, and urinary tract dysfunction, was done. The exclusion criteria were studies not written in English and not including urinary tract dysfunction deep evaluation and description. Studies mentioning general urologic abnormalities without deep description and/or follow-up were not considered. Due to the rarity of the condition, we considered not only papers including pediatric patients, but also papers with pediatric and adult case reports

## 1. Introduction

Wolfram Syndrome (WS) (OMIM 222300) is a rare neurodegenerative disease (RN1290 code) with autosomal recessive inheritance and characterized by juvenile-onset, non-autoimmune diabetes mellitus and later followed by optic atrophy leading to blindness, diabetes insipidus, hearing loss, and other neurological and endocrine dysfunctions [1,2]. It was firstly described by Wolfram and Wagner in 1938 [1]. The acronym DIDMOAD (Diabetes Insipidus, Diabetes Mellitus, Optic Atrophy, and Deafness) includes the main clinical characteristics of the syndrome. More recently, the acronym DIDMOAD has been modified as DIDMOAUD due to increased detection of urinary dysfunction [3]. 

WS is a rare disease with an estimated prevalence varying from 1/770.000 [2] to 1/68.000 of children [4] and with a carrier frequency of one in 354 and a high rate of consanguinity [5]. Several abnormalities in heterozygous relatives of WS patients have been described, like glucose abnormalities, subclinical hearing loss, and neuropsychiatric disturbances including suicidal behavior. 

Mortality is about 65% before age 35 years (age range 25–49) due to central respiratory failure with brainstem atrophy, hypoglycemic coma, status epilepticus, suicide, and renal failure secondary to infections [6].

### 1.1. Clinical Characteristics

According to the International Classification of Diseases (ICD-11), Wolfram Syndrome is categorized as a rare specific diabetes mellitus (subcategory 5A16.1) [7].

Diabetes mellitus (DM), due to insulinopenia secondary to degeneration of β-cell in absence of autoimmunity, is an invariable finding that occurs as the first manifestation, has a non-autoimmune origin, and is insulin requiring. Due to residual endogenous insulin secretion, ketoacidosis is rare and long-term degree of glycometabolic control is better than in type 1 DM, insulin requirement is low, without chronic microangiopathic complications. Previous autoptic studies showed loss of β-cells or atrophy of the islets in the pancreas from WS patients, while the exocrine portion of the gland was normal or with focal areas of fibrosis [5,6]. Immunohistochemical studies of the pancreas in patients with WS demonstrated the absence of cell staining for insulin, but normal cell staining for glucagon, somatostatin, and pancreatic polypeptide, thus indicating a selective β-cell loss with better preservation of the exocrine component of the gland. Therefore, WS associated DM is caused not by a functional defect in β-cells, but by actual β-cell progressive depletion [5,6]. 

Diabetes insipidus (DI) has most commonly a central origin and is usually diagnosed in the second decade, requiring intranasal or oral desmopressin replacement treatment [8,9]. Polyuria and polydipsia in the absence of glycometabolic imbalance are the first symptoms and should induce one to suspect ongoing DI. Neuroradiological findings include the absence of normal T1 hyperintensity in the posterior pituitary gland and functional defects of the paraventricular and supraoptic nuclei with gliosis and atrophy [10]. 

Ophthalmological findings include severe axonal loss and demyelization of the optic nerves, as well as chiasma tract [11], resulting in optic atrophy (OA), usually diagnosed in the first decade of life. Initially asymptomatic, and eventually found during routine eye screening in DM, OA worsens up to reduced visual acuity up to blindness, even at a young age. Retinal thinning detected by optical coherence tomography is a marker of OA progression [12]. Blindness is a consequence of OA, since diabetic retinopathy, as well as other microangiopathic complications related to diabetes mellitus, are almost absent [13]. Cataract, pigmentary retinopathy, and glaucoma, even though extremely rare, have been anecdotally reported in small case series [14]. The pathogenesis of OA could result from the effects of WS mutation on the survival of retinal ganglion cells, leading to anterograde atrophy of retinal axons and shrinkage of the optic nerve [11]. The hypothesis that Wolframin is expressed in glial cells of the optic nerve and in retinal ganglion cells has been tested in cynomolgus monkeys, using affinity-purified antibodies to Wolframin; retinal ganglion cells and optic nerve glial cells were found to be strongly labeled, which suggests that dual dysfunction of Wolframin in the optic nerve and in retinal ganglion cells might explain the progressive OA [15]. 

Sensorineural deafness (SD) is usually diagnosed at a median age of 16 years in 60% of cases [3,16]. Audiometric features include a severe auditory threshold shift, more evident for the medium/high frequencies. SD could be a consequence not only of dysfunction of cochlear neurons and VIII nerve fibers, but also of the central nervous pathways in the brainstem and inferior colliculus [17]. 

Other features of WS include endocrine dysfunctions, in particular primary and secondary hypogonadism, more represented in the male gender, while in females only menstrual abnormalities are frequently encountered [2,8,18]. Anterior pituitary hypofunction seems to have a hypothalamic origin and includes growth hormone deficiency and impaired corticotrophin secretion [8]. Growth velocity and pubertal development need to be carefully followed. Moreover, during a period of stress or infectious diseases, steroid supplementation is recommended. 

A wide spectrum of neurodegenerative abnormalities affecting the central nervous system has been described, including anosmia, ataxia, seizures, nystagmus, gaze palsies, dysarthria, dysphagia, gait impairment, severe psychiatric disturbances, cognitive impairment, central apnea, neurogenic upper airway collapse, and myoclonus [18,19]. Among these complications, neurogenic bladder and urodynamic abnormalities also deserve attention.

#### Genetic Diagnosis

The nuclear gene for WS is mapped on chromosome 4p16.1 [20] and is composed of eight exons: the first is noncoding, the second to the seventh are coding exons, and the eighth is 2.6 kb long-spanning 33.4 kb of genomic DNA. The 3.6-kb mRNA encodes an 890-aminoacid hydrophobic and tetrameric protein named Wolframin (WFS1) [21], composed of nine transmembrane segments and large hydrophilic regions at both termini [22]. WFS1 mRNA is expressed in the pancreas, brain, heart, skeletal muscle, placenta, lung, liver, and kidney. Disease-causing mutations are missense, nonsense, frameshift, and aminoacid insertions or deletions have been reported. WS-associated genotypes include homozygosity or compound heterozygosity, and mutations are located all over the WFS1 gene.

Biochemical studies in cultured cells indicate WFS1 to be an integral, endoglycosidase H-sensitive membrane glycoprotein that primarily localizes in the endoplasmic reticulum. Evidence suggests that WFS1 is either a novel endoplasmic reticulum calcium channel or a regulator of channel activity [23,24]. WFS gene mutations are responsible for endoplasmic reticulum stress-mediated apoptosis with diabetes mellitus and multi-organ failure due to a deficiency of a single protein. It has been demonstrated that WFS1 is a calmodulin (CaM)-binding protein. CaM targets several cellular proteins to provide a wide range of Ca signal transduction [24,25]. 

In 2000, a second locus, named Wolframin 2 (WFS2), was mapped on chromosome 4q22-q24 following the linkage analysis of four consanguineous Jordanian families [26]. WFS2 mutated patients are not affected by diabetes insipidus, but show upper gastrointestinal ulceration and bleeding. The ZCD2-encoded protein, ERIS (Endoplasmic Reticulum Intermembrane Small protein), is also shown to localize to the endoplasmic reticulum (ER), but does not interact directly with Wolframin [27].

Wolfram Syndrome is a devastating disease, involving patients’ and families’ quality of life and life expectancy [28,29]. Since WS is a rare disease, the risk of misdiagnosis has been reported, leading to improper therapy and delay of associated illnesses [30,31]. A correct method to evaluate disease severity is an essential prerequisite for any new treatment protocol [32].

### 1.2. Urinary Tract Dysfunction

Urinary Tract Dysfunctions (UTD) up to end-stage renal disease are a life-threatening complication of WS patients [2]. Noteworthy, end-stage renal disease is reported as one of the most common causes of death among WS patients [2]. The presentation and nature of UTD are controversial and longitudinal evaluation of urinary dysfunction in large case series has not been reported up to now. Moreover, UTD are initially asymptomatic, and if not correctly detected, some symptoms may be erroneously interpreted and ascribed to microangiopathy complicating diabetes mellitus, with misdiagnosis or improper treatment. On the other hand, glycemic variability usually considered the *primum movens* of endothelial damage is rarely reported in WS-related diabetes [33]. UTD have been defined as a late feature of the syndrome, developing in the third decade of life [2]. However, UTD have been also reported in affected adolescents [34]. Involvement of urinary tract occurs in about 90% of affected patients, at a median age of 20 years and with peaks at 13, 21, and 33 years [35]. 

UTD are characterized by a broad spectrum, including bilateral urinary tract dilatation varying from mild hydronephrosis up to megaureter, atonic bladder, and megacystis [3]. High capacity atonic bladder has been described, otherwise low capacity high-pressure bladder with sphincter-detrusor dyssynergia has also been reported [36]. Upper tract dilatation, urinary incontinence, enuresis, post-void residual bladder volume, and recurrent infections are a consequence of neurogenic bladder. 

Conflicting data are reported about UTD pathogenesis: initially, it was attributed to increased urinary output due to diabetes insipidus, or to a functional obstacle in urine output [2]. The observation of UTD in Wolfram Syndrome 2, whose main clinical characteristics include gastrointestinal ulcers due to bleeding tendency, diabetes mellitus, optic atrophy, deafness, hypogonadism, in absence of diabetes insipidus [25,26,37], the different clinical phenotypes of UTD among patients and the deeper knowledge about neurological impairment shed light on its neurodegenerative origin, as primary manifestation [38]. UTD are unlikely to be a consequence of diabetic peripheral neuropathy since it has been reported in young patients with a short duration of hyperglycemia. 

In patients affected by WS, periodical evaluation of urologic and renal tract abnormalities is recommended, including ultrasound, urodynamic examination, and assessment of bladder voiding ability. In case of bladder dysfunction or any other abnormality, periodical urine cultures are mandatory for prompt recognition of infections. Treatment of neurogenic bladder consists of clean-intermittent self-catheterizations or indwelling catheter, and urinary tract infections proper antibiotic treatment.

The aim of this narrative review was to provide an overview of the most important papers regarding renal and urologic involvement in Wolfram Syndrome. 

## 2. Materials and Methods

A comprehensive search of English language articles was performed in the PubMed database (National Library of Medicine, Rockville Pike Bethesda, MD, USA) without using other search filters, except for the publication interval between 1982 and 2020. 

Two authors performed the search, and the keywords were Wolfram syndrome, chronic renal failure, bladder dysfunction, urological aspects, and urinary tract dysfunction. A manual search in the reference lists of the most significant papers was also performed. Studies not written in English and/or not including urinary tract abnormality evaluation were excluded. Each study was screened by title and abstract. For each eligible study, we extracted the following data: author, year of publication, design of the study, population studied, control group (if available), RHI results, RHI outcomes. Due to the rarity of the condition, we considered not only papers including pediatric patients, but also papers with pediatric and adult case reports. Results are summarized in Tables.

## 3. Results

In Table 1, we summarized original articles, and in Table 2, the case reports focused on urological complications in patients with Wolfram Syndrome.

Among the case reports, Piccoli reported a 31 years old male diagnosed as type 1 DM at the age of 7 years, treated with multiple daily insulin injections and persistent satisfactory degree of metabolic control, who 24 years after, showed unexplained increased serum creatinine values with oligoanuria and bladder globus. Occasional dysuria was referred. Optic atrophy was also reported. Ultrasound revealed bilateral dilatation of upper urinary tract ureters associated with bladder distension. Urography showed marked and diffuse and bilateral dilatation of the upper and lower urinary tracts. Urodynamic study revealed detrusor-sphyncter dyssynergia. Self-catheterization was prescribed. Renal involvement was a consequence of neurodegeneration in WS, and despite treatment, ultrasound picture did not improve, and neurological worsening was reported [39].

Nakamura described an adult with DM diagnosed at 6 years of age, loss of vision at 11, and hearing loss at 19. All these symptoms were compatible with WS. An uncommon finding in WS patients, i.e., positivity for β-cell autoimmune markers (Glutamic Acid Decarboxylase and Insulinoma-associated Antigen-2 Antibodies) was observed. At the age of 24, because of reported difficulty in urine output, daily bladder catheterization was prescribed. Urodynamic evaluation showed bladder atony without upper urinary tract anomalies. Brain magnetic resonance showed absence of posterior pituitary signal without cerebellar and brainstem atrophy. Genetic analysis showed WFS1 gene mutation consisting of a homozygous five base pairs (AAGGC) insertion at position 1279 in exon 8, which causes a frameshift at codon 371 leading to premature termination at codon 443 [40]. 

Hasan described a 15 years old boy from consanguineous parents and positive family history for DM, who developed insulin treated hyperglycemia at the age of 9 years, with adequate metabolic control. Two years after the patient referred dysuria, fever, chill, nicturia, and urinary tract infection due to Escherichia Coli was diagnosed. Abdominal ultrasound revealed dilated right pelvi-calyceal system and small left kidney. Urography showed mild dilated right kidney and non-functioning small left kidney. Radionuclide scan by DMSA 99mTc showed small left kidney with poor uptake and right kidney with moderately reduced uptake. Cystoureterography showed smooth wall of bladder and severe bilateral vesico-ureteral reflux up to both calyceal systems with dilatation of the right kidney and both ureters. Grade 5 reflux and neurogenic low-pressure atonic bladder were diagnosed. Prophylactic antibiotic treatment and clean intermittent catheterization were prescribed. Due to poor compliance, recurrent urinary tract infections were reported. Ultrasound showed dilated right kidney and small left kidney. Urological impairment worsened up to chronic renal failure at age of 13 years, requiring twice weekly haemodialysis. Hearing loss and optic atrophy, the main features of WS, were diagnosed at age of 12, after urodynamic impairment. Brain magnetic resonance showed generalized brain atrophy reduced pituitary gland volume and lack of posterior pituitary signal. Due to renal failure water deprivation test aimed to diagnose DI was not done. Noteworthy, the patient showed also upper gastrointestinal bleeding, a typical feature of WFS2. Differently from the previous two cases, urodynamic complications developed 1 year before OA and 2 years after DM diagnosis [41].

Thanos reported three siblings (2 males and 1 female) with WS. All of them showed dilatation of the urinary tract with bilateral hydronephrosis. Hydronephrosis was severe and bilateral in the 39 years old female and in the 31 years old male, while was mild in the 23 years old male. The latter two presented distended-hypotonic bladder. Urological complication have been diagnosed at least 20 years after DM and 10 years after OA [42]. 

Yuca, in 7 siblings affected by WS belonging to a family of 20 subjects, with mean age 10.8 ± 4.4 years and with clinical onset of DM between 2 and 7 years of age, studied urodynamic evaluation and revealed precocious urological complications between 8 and 17 years of age in 4 cases. All of them showed also optic atrophy, hearing loss and DI. In particular, atonic bladder, detrusor weakness and residual urine in the bladder was found in 3 females aged 8, 8.5 and 9 years, respectively, who developed end stage renal failure within 2 years. Neurogenic bladder with residual urine and retrograde renal injury, together with free water loss secondary to DI was responsible for renal failure, requiring haemodialysis. The oldest sibling presented neurogenic bladder at 17 years of age without renal failure [35].

Tekgul described the results of a complete urological evaluation in 14 WS unrelated patients, six girls and eight boys, with a mean age of 13.4 years. Abdominal ultrasonography showed upper tract dilatation in 11 patients, 10 with bilateral hydronephrosis and one with unilateral hydronephrosis, with different grades of ureteral reflux. Urodynamic evaluation showed normal bladder function in only one patient with severe hydronephrosis. In seven patients, low capacity high pressure bladder was observed. Moreover, lower urinary tract was involved in six cases with larger atonic bladder, four with emptying problems, five with low compliance bladder, emptying problems and sphinteric dyssynergia and 2 with hyperreflexic bladder and sphinteric dyssynergia. The severity of hydronephrosis was not related to the nature of bladder dysfunction. Average time after DM clinical onset was 77 months for a low compliant bladder and 84 months for a high compliant bladder [34].

Mozafarpour reported a case series of 27 WS patients who underwent a multidisciplinary evaluation over 10 years. From 1996 to 2005, 12 patients were managed in the pediatric urology center for bladder dysfunction and bilateral hydroureteronephrosis, and underwent urinalysis, urine culture, renal function test, urinary tract ultrasound, voiding cystoureterography, magnetic resonance urography, urodynamic study, and were taught on clean intermittent catheterization (CIC). Among these 12 patients, five females and one male, aged 7–17 years, showed lower urinary tract symptoms and recurrent infections and due to blindness were unable to perform CIC. Therefore, they underwent appendico-vesicostomy according to Mitrofanoff procedure, followed by improved quality of life, reduction of upper tract dilation, degree of hydronephrosis, recurrence of infections and of levels of serum creatinine. In two patients who did not perform regular CIC, grade III bilateral hydroureteronephrosis developed [43].

In another study by Wragg young WS patients, aged 3.2–22.9 years, were evaluated in a multidisciplinary center. Bladder evaluation included non-invasive urodynamic study, bladder capacity, voided volume, post-void residual, and uroflow pattern. Bladder capacity was defined as percentage predicted bladder capacity. Symptoms, bladder behavior and genotyping were also correlated. Normal bladder function was observed in four cases, and abnormal bladder function in the remaining 34. In particular, overactive bladder was found in nine cases and underactive bladder function in the remaining 25. Symptoms were present in only 11 children, all with abnormal bladder function. Daytime urinary incontinence was observed in seven cases, and three were on clean intermittent catheterization. Diurnal incontinence was found in three cases, and nicturia in one. No urinary tract infections were reported. Megacystis, a progression from bladder dysfunction, was observed in older patients, age range 13.9–18.7 years [44].

A longitudinal study was performed by Rove between 2010 and 2016. Thirty-six children, adolescents and young adults with genetically confirmed WS, median age 16.9 years, underwent non-invasive urodynamic testing, including pre- and post-void pelvic ultrasound and uroflowmetry. Symptoms like urinary frequency, urgency and fecal/urinary incontinence were recorded, brain magnetic resonance and complete-validated outcome measures were included in the multidisciplinary evaluation. Functional bladder capacity was decreased in 41% of patients, normal in 65%and increased in 21%, without significant variations across time. At least in one evaluation, 44% of patients showed abnormal uroflowmetry and 54% post-void residual. Two or more brain magnetic resonances were performed in 21 patients, and in 76% of cases deterioration in pons volume was found. Abnormal urodynamic parameters were associated with reduced pons volume [38]. Noteworthy, brainstem and pons host Barrington nucleus, the pontine micturition center [45,46]. 

Lombardo described an extended five-generation pedigree with atonic bladder and enuresis in two girls, aged 15 years and 11 years, respectively. A homozygous 16-bp deletion in exon 8 of WFS1 gene, with a stop codon in position 454 was found in all cases [47].

Yu in 2010 reported a young WS patient with consanguineous ancestors with urological complications diagnosed at 16 years of age. He complained incontinence and frequent micturition; urodynamic study showed nervous bladder. One year later, he progressed to chronic renal dysfunction. DM was diagnosed 5 years after (21 years of age). The author suggests that “WS should be suspected when continuous urinary tracts disorders are present without clear cause” [48].

Zmyslowska evaluated 9 WS patients, mean age 15.4 + 4.9 years. Neurogenic bladder was found in two females, atonic bladder in one male and one female and nocturnal enuresis in another female. The mean age at diagnosis of DM and OA was, respectively, 5.6, 1.8 years and 9.3 years. In the same study, 22 first degree relative with WS were genetically evaluated but clinical manifestations were not reported [49].

Çamtosun described three WS patients with age range 4–17.25 years. Only a female reported urinary incontinence, enuresis, neurogenic bladder and recurrent urinary tract infections. Urinary complications started at 9 years, DM and OA were diagnosed, respectively at 9.5 and 7 years of age. Curiously, DM has been diagnosed after urinary complications [50].

Barrett studied 45 WS patients and showed renal tract abnormalities in 58% of cases, median age 20 years, range 10–44. Dilated renal outflow tracts was found in 26 patients with urinary frequency, incontinence, and recurrent infections. Patients were managed on clean intermittent self-catheterization or an indwelling catheter. Urodynamic studies in four patients showed bladder instability and incomplete bladder emptying. Four others had complete bladder atony [51].

Dryer reported atonia of the efferent urinary tract often with fatal complications in 46% of patients affected by WS [52].

Kinsley in a survey on 68 WS patients reported the complications of urinary tract atony as the most important cause of death together with neurological impairment [6]. 

Chaussenot evaluated WS patients with precocious neurologic impairment (median age 15 years) and reported cognitive impairment in 32% of cases, especially in those who developed neurologic symptoms before 15 years of age [18]. 

In our case report in WS patients, we observed a double left district in a girl aged 11 years, carrying the frameshift mutation c.2106_2113del8nt F646fs708X [53] andleft-obstructive hydronephrosis at the pyeloureteral junction in a boy carrying homozygous mutation at the nucleotide c.1362_1377del16. Brain nuclear magnetic resonance revealed brain stem, cerebellum, medulla, and pons atrophy (Figure 1) [54]. 

Involvement of the central nervous system is considered the main causative factor for multiorgan impairment reported in WS. 

## 4. Discussion

Results from several case reports and from multidisciplinary centers reported high frequency of UTD. Therefore, this comorbidity and its severe long term consequences should be considered a precocious complication of WS and screening and prompt treatment are mandatory. In fact, based on the review of Kumar, the acronym DIDMOAD has been modified as DIDMOAUD due to increased frequency of urinary dysfunction, not only in adulthood as reported by Barrett [2], but also in young patients, as reported by Tegkul [3,34].

Patients with clinical WS should be followed in a multidisciplinary center with specialized knowledge for correct genetic diagnosis, familiar counseling, regular and extensive clinical work up, since every organ and system can be affected by neurodegeneration. 

Neurodegenerative disorders with DM requiring differential diagnosis with WS include Alstrom Syndrome, Bardet Biedl Syndrome, Friedrich ataxia, Thiamine-responsive megaloblastic anemia, Myotonic dystrophy type 1, Kearns Sayre Syndrome [55]. In particular, Bardet Biedl Syndrome is characterized by renal dysfunction, and Friederich ataxia may include bladder dysfunction [55].

Neuroradiological follow-up is recommended, since early brain vulnerability reported in young patients with WS can be considered a causative factor for the development and progression of multiorgan failure [46]. As regards neuroradiological aspects, it has been reported widespread brain atrophy in late stage WS [10]. However, it is important to define when these abnormalities start, and whether there is specific involvement across brain regions and tissues. To this purpose, the Washington University Wolfram Study Group performed brain magnetic resonance with multiple imaging modalities in a cohort of children and young adult patients with clinical and genetic diagnosis of WS [46]. Patients underwent cognitive and behavioral testing. Results were compared to type 1 DM patients and to healthy control group. Patients with WS group had intact cognition, significant anxiety and depression, and gait abnormalities. Compared to healthy and type 1 DM patients, the WS group had smaller intracranial volume and preferentially affected gray matter volume and white matter microstructural integrity in the brainstem, cerebellum and optic radiations. Different degrees of abnormalities were detected in even the youngest patients even with milder symptoms, and some measures did not follow the typical age-dependent developmental trajectory [46]. These results confirm smaller intracranial volume with specific abnormalities in the brainstem and cerebellum in WS, even before or at the earliest stage of clinical symptoms. This pattern of abnormalities suggests that WFS1 gene alteration severely impacts on early brain development, in addition to later neurodegenerative effects [46].

In patients affected by WS periodical evaluation of urologic and renal tract abnormalities is strongly recommended, including ultrasound, urodynamic examination and assessment of bladder voiding ability. The presentation and nature of UTD are controversial and longitudinal evaluation of urinary dysfunction in large case series has not been reported up to now. In case of bladder dysfunction or any other abnormality, periodical urine cultures are mandatory for prompt recognition of infections. Treatment of neurogenic bladder consists of clean-intermittent self-catheterizations or indwelling catheter, and urinary tract infections proper antibiotic treatment. A prompt and correct diagnosis of WS and its related diseases is mandatory to prevent complications and precociously start appropriate treatment [56]. 

Currently, the main objective for WS management is to stop the progression of associated diseases and replacement of damaged tissues, in particular β-cells and retinal cells [7]. At present, specific and effective therapy is not available, and drugs aimed to maintenance of endothelial reticulum homeostasis, calcium homeostasis redox regulation and protein folding represent the future options [56]. Among therapeutic options, chemical chaperones, a class of molecule aimed to protein folding in ER, seem to protect β-cell as well as neuronal cells from death by reducing ER stress, even decelerating neurodegeneration [57]. 

Other promising therapeutic options include dantrolene, which suppresses calcium efflux from ER with subsequent preservation of both β-cells and neuronal cells [58].

Our narrative review has some limitations. First, the rarity of the syndrome, and the subsequently reduced availability of original papers with full description including urinary tract dysfunctions besides the usual clinical phenotype of the syndrome, i.e., diabetes mellitus, diabetes insipidus, optic atrophy and deafness. Moreover, follow up studies regarding treatment and long term consequences are sometimes limited to a low period, and related to a small case series. A comparison of urologic impairment in WS and other forms of diabetes mellitus is lacking.

Longitudinal studies on broader case series and international registries for this rare disease will increase the knowledge and the natural history of WS, and develop new opportunities for intervention and prevention for patients’ and families’ benefit [59,60].

## 5. Conclusions

WS is a neurodegenerative, devastating disease involving several organs and systems, requiring prompt diagnosis and adequate treatment, aimed to prevent the progression of associated diseases. Even if diabetes mellitus and optic atrophy are the main clinical characteristics, especially in the pediatric age group, several reports showed precocious renal and urinary tract involvement, leading to increased morbidity and mortality. 

Pediatricians who take care of affected patients should be aware of the importance of periodical screening of associated diseases, for prompt diagnosis and treatment.

Adequate care of WS patients since pediatric age is the cornerstone for promising future treatment aimed to restore cell function, increase life-long expectancy and improve quality of life.

## Figures and Tables

**Figure 1 ijerph-18-11994-f001:**
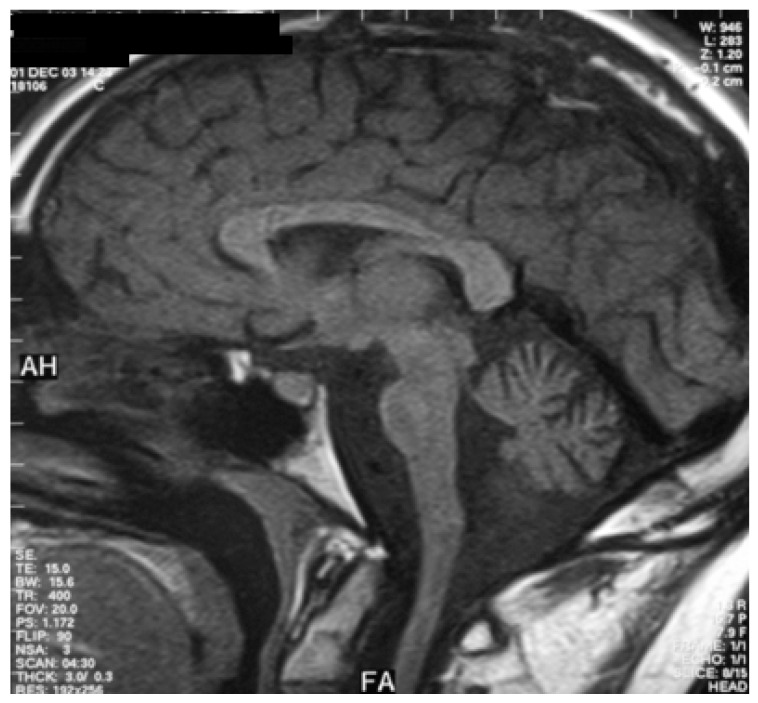
Brain magnetic resonance showing pons and cerebellum hypoplasia (personal observation [54].

**Table 1 ijerph-18-11994-t001:** Overview of the original articles reporting on renal and urologic involvement in Wolfram Syndrome.

Author, Year	Characteristics of Population at Time of Report Description	Age at Diagnosis of Diabetes Mellitus	Age at Diagnosis of Optic Atrophy	Age at Diagnosis of Urological Complications	Urological Complications	Geographic Origin
Thanos, 1992	3 siblings from a Greek family [M/F = 2/1] Average age 31 years Age range 23–39 years	Average age 5.7 years Age range 5–7 years	Average age 16 years Age range 14–20 years	Female 1: 39 years Male 1: 23 years Male 2: 31 years	N = 3 Dilatation of urinary tract Female 1 Severe bilateral hydronephrosis with dilated ureters and distended bladder, Hypotonic bladder Male 1 Mild bilateral hydronephrosis with dilated ureters and distended bladder, Hypotonic bladder Male 2 Severe bilateral hydronephrosis with no other alterations	Greece
Kinsley, 1995	Medical record of 68 patients with Wolfram syndrome from 44 families in 23 states of U.S.A. [M/F = 35/33]	Median age 6.5 years Mean age 8.2 years Age range 1–26 years	Median age 12.6 years Mean age 13.1 years Age range 6–30 years	Dilated Neurogenic bladder: Median age 15.2 years Mean age 17.4 years Age range 7–41 years	N = 37 Dilated Neurogenic bladder N = 27/37 hydronephrosis with dilated ureters N = 21/37 Impaired renal function (elevated urea nitrogen and creatinine) N = 31/37 episodic of chronic cystitis or pyelonephritis N = 5/37 died from end-stage renal disease or consequence of urinary tract atony	U.S.A.
Barret, 1995	45 patients withWolfram Syndrome studied in UK. Only 35 patients were alive during the study. 26 had renal tract abnormalities Median age 29 years Age range 5–46 years	Median age 6 years Age range 3 weeks-16 years	Median age 11 years Age range 6 weeks-19 years	Median age 20 years Age range 10–44 years	N = 26 Dilated renal outflow tract, urinary frequency, incontinence, recurrentinfections N = 4/26 Bladder instability and incomplete bladder emptying N = 4/26 Bladder atony	U.K.
Tekgul, 1999	14 patients with Wolfram Syndrome who underwent complete urological evaluation with ultrasonography and urodynamic [M/F = 8/6] Mean age 13.4 years Age range 7–19 years	Not reported	Not reported	Not reported Note: Average time after diabetes mellitus diagnosis - 77 months for low compliant bladder - 84 months for high compliant bladder	N = 11/14Upper tract dilatation N = 10/11 bilateral hydronephrosis N = 1/11unilateral hydronephrosis [n = 1 grade 1; n = 5 grade 2; n = 2 grade 3; n = 3 grade4] [Grade 3 and 4 have ureteral tortuosity and kinking] N = 3/14Unilateral vesicoureteral reflux N = 6/14 Large atonic bladder (n = 4 with emptying problems) N = 5/14 Low compliant bladder with low capacity (n = 2 with emptying problems and sphynteric dyssynergia, n = 1 with emptying problems) N = 2/14Hyperreflexic bladder with sphynteric dyssynergia	Turkey
Lombardo, 2005	7 related patients sharing two common ancestors in a family from a small isolated town in Nebrodi Mountains of Sicily [M/F = 3/4] Age range 11–17 years	Age range 3–6 years	Age range 10–14 years	Age range 10–11 years	N = 1Acute renal failure, atonic distended bladder, bilateral Hydronephrosys with dilated ureters N = 1Enuresis	Italy
Zmyslowska,2011	9 Polish patients not related [M/F = 1/8] Mean age 15.4 ± 4.9 years	Mean age 5.6 ± 1.8 years Age range 3.8–8.7 years	Mean age 9.3 years Age range 5–20 years	Not reported	N = 1 Nocturnal enuresis N = 2 Neurogenic bladder N = 2 Atonic bladder	Poland
Yuca, 2011	7 siblings from a Turkish family; [M/F = 4/3] Mean age 10.8 ± 4.4 years Age Range 6–19 years	N = 7 Mean age 4.5 ± 1.9 Range 2–7 years	N = 6 Mean age 6.3 ± 1.3 years; range 6–9 years N = 1 not diagnosed	N = 4 Age range 8–17 years N = 3 Not diagnosed	N = 3 Neurogenic bladder(8–8.5–9 years) + Renal failure(9–10–11.5 years); Renal failure was observed two years after detection of neurogenic bladder N = 1Neurogenic bladder (17 years) Urethral dilatation, vesico-ureteral reflux, recurrent urinary tract infection	Turkey
Çamtosun, 2015	3 unrelated patients [M/F = 1/2] Age range 4–17.25 years	Male 1: 1.75 years Female 1: 3 years Female 2: 9.5 years Age range: 1.75–9.5	Male 1: Not diagnosed Female 1: 15.5 years Female 2: 7years Age range: 1.75–9.5	Female 2: 9 years	Male 1: Not reported Female 1: Not reported Female 2: Urinary incontinence, enuresis, recurrent urinary tract infections, neurogenic bladder	Turkey
Mozafarpour 2015	27 patients with Wolfram Syndrome from Children’s hospital in Teheran, Iran. 12 managed in the pediatric urology Center for urinary complications. 6 had severe lower urinary tract symptoms. They were from two families with consanguineous parents [M/F = 5/1] Age not reported	Not reported	Not reported	N = 6/12 not reported N = 6/12 Age range 7–17 years (Patients with urinary complications treated with appendico-vesicostomy)	N = 12 Bilateral hydroureteronephrosis and advanced bladder dysfunction N = 6/12 Severe lower urinary tract symptoms and recurrent urinary tract infections. All underwent appendico-vesicostomy. All progressed to end stage renal failure	Iran
Wragg, 2018	40 patients with Wolfram syndrome from Birmingham Children’s Hospital. 38 had undergone non invasive urodynamics to evaluate bladder function Median age 14 years Range 3.24–22.9 years	Not reported	Not reported	N = 9 Over active bladder, age range 9.3–17.9 years N = 25 Under active bladder age range 12.8–21.0 years Megacystisrange 13.9–18.7 Patients with megacystis were older than patients without megacystis (range 9.7–16.1 years). Patients with bladder dysfunction were older than patients with normal bladder function (range 4–8–11.5 years)	N = 4 Normal bladder function N = 34 Bladder dysfunction (n = 9 overactive bladder; n = 25 underactive bladder) None reported urinary tract infections Megacystis represents a progression from bladder dysfunction. N = 11/34 Reported symptoms (10 urinary incontinence, 1 nicturia)	U.K.
Rove, 2018	36 patients fromWashington University Research Clinic. All underwent non invasive urodynamic testing [M/F = 13/23] Mean age 16.9 years Range 7–30 years	Not reported	Not reported	Not reported	N = 14 Decreased functional bladder capacity N = 7 Increased bladder capacity	U.S.A.

**Table 2 ijerph-18-11994-t002:** Overview of the case reports on renal and urologic involvement in Wolfram Syndrome.

Author, Year	Characteristics of Population at Time of Report orCaseReport Description	Age at Diagnosis of Diabetes Mellitus	Age at Diagnosis of Optic Atrophy	Age at Diagnosis of Urological Complications	Urological Complications	Geographic Origin
Dreyer. 1982	2 German siblings with Wolfram Syndrome with different clinical manifestations. These cases are compared with 98 patients reported in literature Female: 19 years old Male: 5 years old	Female: 7 years Male: 3 years 98 patients from literature, average age 7 years	Female: 10 years Male: 5 years 96/98 patients from literature, average age 11 years	Female: not reported Among 98 patients from literature: not reported	Female: hydronephrosis with dilated ureters and distended bladder Male: not reported Among 98 patients from literature: renal failure as consequence of urinary tract atony and repeated urinary infections	Germany
Hasan, 2000	Male, Age 15 years	9 years	12 years	11 years	Dilated right kidney, non-functioning small left kidney, Bilateral vesico-ureteral reflux at grade 5, Atonic Bladder At 13 years chronic renal failure	Jordan
Piccoli, 2003;	Male, Age 31 years	7 years	Not reported	26 years (supposed)	Bilateral dilatation of upper and lower urinary tract, bladder globus with detrusor-sphynteric dyssynergy	Italy
Nakamura, 2005	Male, Age 47	6 years	11 Years	24 years	At 24 years difficulty in urine output Atonic bladder	Japan
Yu, 2010	One Chinese male, with consanguineous ancestors, who underwent urodynamic study and Kidney CT scan Age 26 years	21 years	Not Reported	16 years	Nervous bladder with incontinence and chronic renal dysfunction	China

## Data Availability

This is a narrative review, data described have been collected by the articles in the reference section. No personal data, except for the figure mentioned, are present.

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
