# Peer review of "Urinary Tract Involvement in Wolfram Syndrome: A Narrative Review"

_ijerph, 2021, doi:10.3390/ijerph182211994_

Round 1
Reviewer 1 Report
The following review aims to organize published reports in the literature related urinary tract dysfunction in Wolfram Syndrome patients.
Abstract: please remove any repetitive language that is also found in the manuscript body. Several sentences in the abstract are directly copied from the introduction and discussion. The abstract should be concise and describe the problem, question, methods, and any major findings.
Introduction:
Lines 72-73: grammatical and syntax errors
Lines 123 -124: remove "has". What does 2000 refer to? year?, publication?
Methods:
Lines 171-172: How did the authors determine the importance of the papers selected? This approach seems bias and could undermine the purpose of a review.
Results:
Table 1: What does "we mainly focused on" mean in reference to the reports listed in table 1? It would be helpful if more than one table is provided to accurately represent the different types of publications reviewed.
As currently written, it is unclear if the authors described all the published articles listed in the table (see lines 189-331).
The authors are not consistent when describing certain case reports. For example, authors give the name of one author or provide the designation "et al." for others.
Are the references for each study described listed at the end of each paragraph? Either use the citation to describe the case at the start or cite the reference at the end of the paragraph, not both.
What value does providing the country of origin for each case provide?
Did the authors describe all the case reports listed in the table? If not, how were the case reports described selected and not others?
The table contains studies not mentioned in the text.
Figure 1. What does personal observation mean? (lines 333-334)
Discussion:
Line 352: spelling and grammar error
Consider revising the discussion to address the aim of the paper to review UTD in WS more specifically, rather than broadly as currently written.
Conclusion: Although the authors provide a useful collection of published data on Wolfram Syndrome, a systematic re-write of results and discussion could greatly improve the manuscript.
Author Response
Dear Sir
Thanks for your observations. We answer as follows:
Abstract: please remove any repetitive language that is also found in the manuscript body. Several sentences in the abstract are directly copied from the introduction and discussion. The abstract should be concise and describe the problem, question, methods, and any major findings. The abstract has been modified.
Lines 72-73: grammatical and syntax errors: amended
Lines 123 -124: remove "has". removed What does 2000 refer to? It refers to calendar year of publication
Methods:
Lines 171-172: How did the authors determine the importance of the papers selected? This approach seems bias and could undermine the purpose of a review.
Our aim was to perform a search on Pubmed including Wolfram Syndrome and one or more following terms: chronic renal failure, bladder dysfunction, urological aspects, urinary tract dysfunction. The exclusion criteria were studies not written in English and not including urinary tract dysfunction deep evaluation and description. Studies mentioning general urologic abnormalities without deep description and/or follow-up were not considered. We also focused on original articles and case reports; therefore the original table has been divided in two: one for original articles, one for case report.
Results:
Table 1: What does "we mainly focused on" mean in reference to the reports listed in table 1? It would be helpful if more than one table is provided to accurately represent the different types of publications reviewed. Done
As currently written, it is unclear if the authors described all the published articles listed in the table (see lines 189-331). Yes, we checked the text ant the tables.
The authors are not consistent when describing certain case reports. For example, authors give the name of one author or provide the designation "et al." for others. Amended
Are the references for each study described listed at the end of each paragraph? Yes
Either use the citation to describe the case at the start or cite the reference at the end of the paragraph, not both.
What value does providing the country of origin for each case provide? Due to rarity of the disease and the different clinical phenotypes, we considered to specifcy the country of origin for a more accurate report.
Did the authors describe all the case reports listed in the table? Yes
If not, how were the case reports described selected and not others?
The table contains studies not mentioned in the text. We performed a double check and added the name of two authors previolusly not mentioned (i.e. Yuca , ref. 35 and Lombardo, ref 37)
Figure 1. What does personal observation mean? (lines 333-334) It reports the image of our patient already published, ref. n. 54. The parenthesis has been added at the end of the sentence.
Discussion:
Line 352: spelling and grammar error Amended
Consider revising the discussion to address the aim of the paper to review UTD in WS more specifically, rather than broadly as currently written. The discussion has been modified including limitations of the review.
Conclusion: Although the authors provide a useful collection of published data on Wolfram Syndrome, a systematic re-write of results and discussion could greatly improve the manuscript: The results section and the discussion have been modified.
Reviewer 2 Report
Although the topic discussed in this paper entitled: "Urinary tract involvement in Wolfram syndrome: a narrative review" is of interest to readers, and both the scientific rigor together with the scientific content quality are relevant, some revisions are needed before publication.
Line 171-172: The aim should be moved at the end of the Introduction section
Line 184-185: Please summarize the total number of articles found, the number of articles included in the review, and the exclusion criteria
Result section: when an article is cited please include "et al" and publication year after the author's name (i.e. line 189-199-210 etc..)
Discussion section: line 370-372 This period should be moved to result section
Author Response
Dear Sir
thanks for your suggestions. Here enclosed our answers
Line 171-172: The aim should be moved at the end of the Introduction section. Done
Line 184-185: Please summarize the total number of articles found, the number of articles included in the review, and the exclusion criteria. Being a narrative review, we performed a search on Pubmed including Wolfram Syndrome and one or more following terms: chronic renal failure, bladder dysfunction, urological aspects, urinary tract dysfunction. The exclusion criteria were studies not written in English and not including urinary tract dysfunction deep evaluation and description. Studies mentioning general urologic abnormalities without deep description and/or follow-up were not considered. Since the aim of our study was to perform a narrative review only, if the reviewer would suggest to include other papers, we
Result section: when an article is cited please include "et al" and publication year after the author's name (i.e. line 189-199-210 etc..). "et al" after some authors' names has been deleted.
Discussion section: line 370-372 This period should be moved to result section. Done
We hope that our answers will be in accordance with your aims.
Best regards
Giuseppe d'Annunzio and co-authors
Reviewer 3 Report
The authors conducted a literature review of urinary tract disorders in patients with Wolfram Syndrome (WS). This is an important review, showing that the development of urinary dysfunction is one of the main symptoms of WS and that those involved in the management of WS patients need to pay attention to it. This review is well organized and potentially interesting; however, I have some concerns as below:
- Although the authors use the term "urinary dysfunction", most reported cases are neurogenic bladder dysfunction, and urinary tract abnormalities such as hydronephrosis and vesicoureteral reflux appear to be a concomitant symptom of this neurogenic bladder rather than congenital. In this case, the authors have already explained at the introduction that neurological dysfunction is one of the characters of WS. Is neurogenic bladder not included in one of these neurological dysfunctions?
- Urinary tract abnormalities in some syndromes such as BOR syndrome, CHARGE syndrome, Kalman syndrome, and Alagille syndrome, and so on are often detected by maternal ultrasound during the fetal period and are so-called "congenital renal urinary tract malformations (CAKUT)". How about WS patients? Do urinary tract abnormalities occur acquiredly or are they simply delayed in diagnosis? Although the authors state that WS has a high complication rate for renal urinary tract abnormalities, but it is necessary to consider why it was not previously included as one of the major symptoms of WS. Authors need to add their views to the discussion.
- The results section summarizes the content of the case reports in a table, which is very verbose because the contents are repeated in text. The results section should be simplified by omitting the infromation already in the table.
Author Response
Dear Sir
thanks for your suggestions.
As regards first observation, urinary dysfunction with different clinical manifestations is a main feature of Wolfram Syndrome, initially considered as a late complication. Thanks to increased knowledge and diagnostic procedures, expecially neuroradiological ones, the urinary dysfunction can be detected also in adolescence. By the way we think it is quite difficult to establish if urinary dysfunction is congenital or acquired. To our knowledge, only one case report described perinatal Wolfram Syndrome, characterized by diabetes insipidus and optic atrophy. The paper does not mention urological evaluation (Ghirardello F et al; IJP 2014;40:76).
As regards the second observation, according to the literature urologic abnormalities are frequent, otherwise not universally detectable. In fact, WS clinical diagnosis is based on occurence of non autoimmune diabetes mellitus and optic atrophy, at least for pediatric age group. To our knowledge only one prenatal diagnosis of WS by transabdominal chorionic villous sample was published (Domenech E et al; Prenatal Diagnosis 2004; 24:787-789).
As regards the third observation, the results section has been shortened.
We hope that our answers will be in accordance with your aims.
Best regards
Giuseppe d'Annunzio and co-authors
Round 2
Reviewer 1 Report
Suitable. Please check for continued minor grammatical errors.
Reviewer 3 Report
The revised version of the manuscript has been improved by making some changes and additions.
I have no additional comments.